# Is Platinum a Real Store of Wealth?

**Marek Vochozka** *, **Andrea Bláhová** and **Zuzana Rowland** *

Research Department of Economics and Natural Resources Management, Institute of Technology and Business in České Budějovice, 37001 České Budějovice, Czech Republic
* Correspondence: vochozka@mail.vstecb.cz (M.V.); rowland@mail.vstecb.cz (Z.R.); Tel.: +420-725-007-337 (M.V.)

**Abstract:** The research goal is to determine whether platinum can be seen as a good investment. For this purpose, content analysis of documents and deep learning neural networks with recurrent neural network were used. The results show that it pays for a koruna investor (a person holding their wealth in Czech koruna) to preserve their wealth physically in the form of a precious metal—specifically, platinum. The research confirms that platinum is a store of value but also a koruna investor's wealth multiplier. This can be due to its rare occurrence in nature, but also to its unique use in manufacturing. A research limitation is the period for which the data were used. The finding that platinum is a store of value, as well as a wealth multiplier, can thus be concretized when using the data for a five-year period. It shall also be added that no turbulent changes are anticipated (such as interruption of platinum supply, unexpected government regulation of trade, etc.).

**Keywords:** platinum price; store of wealth; neural networks; koruna investor; wealth

## 1. Introduction

Platinum is a very heavy and precious metal used in chemical, glass, and pharmaceutical industries. Measurable quantities of platinum, palladium, and rhodium even in remote areas of the Earth prove global pollution by these metals, mostly from the catalytic converters of modern vehicles (other uses for the resources include jewelry manufacturing, the chemical industry, and cancer medicines) (Rinkovec 2019). The world's leading producers of PGE (platinum group elements) are Russia and South Africa (O'Connor and Alexandrova 2021).

Platinum metals enter the environment mainly by means of the global use of catalytic converters in cars (Komendova 2020). Platinum is the most suitable catalyst for accelerating electrochemical reactions in fuel cells. Despite its high price, platinum is used as a cheaper, sustainable, and ecological catalyst for manufacturing (Karim et al. 2019). Platinum as a key catalyst material is important for global green transition both because of its current main use in catalytic converters and its increasing use in newly emerging and renewable energy technologies, such as fuel cells and electrolytic cells (Rasmussen et al. 2019).

This metal can be applied in medicine as well. Platinum compounds represent one of the great successes of metals in medicine (Tylkowski et al. 2018). As one of the important metal-based drugs, it is used in the treatment of solid malignant tumors. Despite serious side effects and acquired drug resistances that have limited its clinical application, platinum shows strong inhibitory effects on a wide range of tumors (Xiao et al. 2020). Platinum-based chemotherapy is the standard of medical treatment for a wide range of cancer types. Platinum-based treatment, including cisplatin, carboplatin, oxaliplatin, or their combination, is used for treatment of various childhood malign tumors. Unfortunately, one of the most serious adverse effects is hearing loss or ototoxicity (van As et al. 2018).

To a lesser extent, platinum is used for making jewelry. The necessary properties of metals used for making jewelry are high hardness and the associated scratch resistance (Houghton and Greer 2021). Platinum jewelry is manufactured using an investment casting process (Chomsaeng et al. 2021).

As such, platinum has a wide range of applications, from automotive through medicine to jewelry making. In addition, it is often used as an investment asset. This is historically given mainly by its rare occurrence in the nature. However, the question is whether the current demand for this precious metal is driven by its necessity (e.g., in automotive, in car catalytic converters) or the effort to allocate the wealth of an individual or a company in an asset capable of keeping its effect. From the investment perspective, the reason is also that the yield of platinum in the commodity markets has a high leverage effect (Bao 2020). On the other hand, Cohen (2022) confirms that daily returns are negatively correlated to the returns on the preceding day. It is thus a question of how to understand and subsequently predict platinum prices.

The gold–platinum (GP) price ratio shows persistent differences in risk and is a proxy for an important economic state variable (Huang and Kilic 2019; Brabenec et al. 2020). The price of platinum is quite high; therefore, cheaper substitutes are sought. Currently, platinum is the best catalyst for the reaction of hydrogen formation; however, its high price significantly restricts its large-scale use. Due to high activity, conductivity, resistance, and low price, a silver catalyst can be considered a promising alternative to commercial platinum on carbon for industrial applications (Li et al. 2019). The monetary policy rate and the production index are important for predicting bubbles in industrial metal prices (Ozgur et al. 2021).

In essence, it can be confirmed that the price of platinum is influenced, determined, or predicted by a number of indicators, including both financial and non-financial ones; it can be the economic cycle of the global economy and many others. A frequently mentioned factor is, e.g., the correlation of platinum and oil prices (Gupta et al. 2021) or, more precisely, platinum and oil prices after a crisis (Aruga et al. 2020). The indicators can be used as a predictive tool; however, it is not possible to consider all predictors of platinum prices, as they are probably too many; moreover, the relationships between these predictors cannot be clearly identified. The predictors include many unquantifiable items, such as sentiment (Maghyereh and Abdoh 2022).

The goal of the paper is to determine whether platinum can be seen as a suitable investment. This is suggested by its scarcity and its practical application. Moreover, it can be accumulated, separated, and thus traded. The application of platinum in technology and medicine can play an interesting role in an investor's decision over whether platinum is a suitable investment.

To see any commodity as an investment, certain characteristics are required. Therefore, the following research questions have been formulated:

RQ1: Is platinum a store of value?
RQ2: Is platinum an investor's wealth multiplier?

Investors need enough time to be able to work with the commodity on the market. Therefore, platinum must also show suitable physical and chemical properties, on which trading techniques are based:

RQ3: What are the properties of platinum as a precious metal?
RQ4: How is platinum traded on commodity markets?

Moreover, potential investors need to be sure they will be able to buy a sufficient amount of a given commodity, which depends on trade technology.

## 2. Literary Research

Serafini and Reid (2019) describe multimodal content analyses, a variant of qualitative content analysis based on previous iterations of qualitative content analysis, interpretive research designs, deductive and inductive reasoning, qualitative data collection and methods of analysis, and theories of multimodality for the conceptualizing and analysis of selected multimodal phenomena corpus. Moreover, an analysis of selected commercial wine labels is presented as an example of multimodal content analysis that would lead to

further research and open up a dialogue concerning potential advantages and challenges for researching multimodal phenomena.

The emergence of social networking sites opened up new opportunities for content analytics, which can be used to analyze user-generated content from a variety of sources. Yet companies are investing millions of dollars in content analysis, often known as sentiment analysis. The overall conclusion is that automated qualitative analysis depends on the accuracy of the tool used, which can be checked by manual qualitative analysis (Anastasiei and Georgescu 2020).

Content analysis is aimed at identifying generative relationships in the field of instruction and to arrange the content in a way that maximally supports generative teaching. Slocum and Rolf (2021) conclude that content analysis is a basis on which generative instruction is built and instructional designers could create more effective, efficient, and powerful programs if they explicitly and in detail attended to content analysis.

Shih (2018) develops a detailed interpretation of sports video analysis with respect to content by examining the findings from research on content structure in various scenarios. Specifically, the authors focus on video content analysis techniques used in sports broadcasts in the last decade in terms of fundamentals and general overview, hierarchical model of content, trends, and challenges.

Rio et al. (2021) discuss the finding of an explanatory content analysis concerning the offender population by analyzing the literature from the years 2008–2018 in 23 American Counseling Association (ACA) and ACA-affiliated counseling journals. The study discussed the method of descriptive content analysis, showing that between 2006–2008, a limited number of dissertations were written, mostly diploma theses at public universities, where the regions of Istanbul, east Marmara, Aegean, and western Anatolia were used as samples (Kosar 2020).

The core of the most common content analysis is the frequency distribution of individual words. The study by Dicle and Dicle (2018) presents a community-contribute command "wordfreq" to process content (online and local content), and for the preparation of frequency distribution of individual words.

The model of autoregressive integrated moving average (ARIMA) shows better performance when predicting short-term trends since it considers the dependence of time series and the interference of stochastic volatility. Analysis performed by Yan and Chen (2017) contributes to the development of current knowledge on predicting the number of application users in the communication market and brings new ideas to increase the market share for communication operators.

Xu and Qin (2021) propose a new hybrid model combining the models of autoregressive integrated moving average (ARIMA) and regression tree (RT) for ITS. Based on experimental analysis, it was found that the performance of the proposed ARIMA-RT model is significantly better than other competitors, mainly in predicting nonlinear ITS, which suggests that ARIMA-RT is able to capture nonlinear ITS on stock markets.

Munim et al. (2019) analyze predictions of bitcoin prices using the models of autoregressive integrated moving average (ARIMA) and neural network autoregression (NNAR). Despite the sophisticated NNAR, the study confirms the enduring power of volatile bitcoin price prediction.

A combined forecasting method based on the ARIMA model and long-short term memory neural network (LSTM) is proposed to improve the accuracy of sales forecasts for manufacturing companies. The combined model shows a higher predictive accuracy and a wider range of applicability (Han 2020). The mechanism, model, and empirical research methods of the changing RMB exchange rate for the competitive advantages of international trade were set on the basis of the autoregressive moving average of time series. The results of empirical analysis confirmed the existence of a necessary connection between RMB appreciation and the persistent surplus of the Chinese balance of trade (Tian 2019).

Bitcoin is one of the most popular cryptocurrencies in the world. Currently, it is attracting the attention of researchers. In the study by Nguyen and Le (2019), ARIMA

(autoregressive integrated moving average) and machine learning algorithms are implemented to predict the closing prices of bitcoin for the next day. The results of the experiment show that hybrid methods improved the accuracy of predictions using RMSE and MAPE (Nguyen and Le 2019).

Vochozka et al. (2020) aim to present a methodology that deals with considering seasonal fluctuations in time-series smoothing using artificial neural networks. They used an example of the balance of trade between the Czech Republic and the People's Republic of China. The benefit of this work is the involvement of variables that are able to predict possible seasonal fluctuations in the trend of time series when using artificial neural networks (Vochozka et al. 2020).

Vochozka et al. (2019) propose a methodology for considering seasonal fluctuation in time-series smoothing using artificial neural networks on the example of the euro and Chinese yuan. Before the experiment, it seemed it was not necessary to include categorical variables in the calculation. However, the results showed that other variables in the form of year, month, day in the month, and day in the week in which the value was measured brought better accuracy and order in time-series smoothing (Vochozka et al. 2019).

Ghazi (2018) examined heavy metals contamination in soils, which can be used to determine the degree of contamination of the environment using new techniques in dependence on the design of backpropagation of neural networks as an alternative precision technique. The results presented in the study show that the proposed networks could be used successfully for a fast and precise estimate of heavy metals concentration.

Vrbka et al. (2020) examined the effect of seasonal fluctuations on smoothed time series using artificial neural networks on the example of imports from the People's Republic of China (PRC) to the USA. Although it seemed before the experiment that there was no reason for including a categorical variable that would reflect seasonal fluctuations in the USA–PRC imports, the assumption turned out to be wrong.

In numerical studies, the proposed method is applied for sensitivity analysis of oil and gold price time series. The results of FSA suggest that oil prices are highly dependent on the inflation rate, dollar index, and market index, while the level of OPEC production and price of gold have a low impact. Moreover, when modeling gold prices, the highest sensitivity is obtained from silver prices, while gold demand is a function of the market index and inflation rate (Lotfi 2019).

Additive manufacturing (AM) of metals is currently a highly discussed area of research in terms of materials processing and manufacturing because of promises of shorter lead time, increased flexibility of design, and location-specific process control. Donegan et al. (2020) develop a method to reduce the overall complexity of a representation of the AM processing space using the techniques of time-series analysis and dimensionality reduction. Lasheras et al. (2022) introduced a methodology for predicting gold prices, which uses the values of this price in the preceding months, as well as values of other metals, such as potash, copper, lead, tin, nickel, aluminum, iron ore, zinc, platinum, and silver, as input information. The proposed methodology is based on the decomposition of each time series in their trend, seasonal, and random components and the application of information on trends as independent variables in a multivariate adaptive regression splines model. In the month-on-month prediction model, the prediction of gold prices between October 2019 and September 2020 showed a mean absolute deviation (MAD) of 67.6022, medium square error (MSE) of 9403.1882, root mean square error (RMSE) of 96.9700, and mean absolute percentage error (MAPE) of 3.8803%.

Plitnick et al. (2018) combine the decomposition of time series and the regression model of time series for predicting floods on the Mohawk River in Schenectady, New York. The results show that the use of time lag of variables in the model of time series more than doubled the accuracy of prediction compared to using raw data only.

Regression algorithms are based on various regression models, i.e., linear regressions, non-linear regressions, multiple regressions, logistic regressions, and probabilistic regressions. The methodology proposed by Sagar et al. (2019) discussed the prediction of

time-series datasets with improved parameters. Afterward, the values of coefficients for the regression model are identified so that they fit the regression model (Sagar et al. 2019).

A functional (lagged) regression model of time series includes the regression of scalar-response time series on a time series of regressors consisting of the sequence of random functions. Rubin and Panaretors (2020) consider the so-called sparse observations, when only a relatively small number of measurement locations are observed, which are different for each curve. The performance and implementation of the methods are illustrated using a simulation study and analysis of meteorological data.

To improve the accuracy and reliability of the prediction of the long-term financial trend, a financial time-series prediction framework is proposed that combines empirical mode decomposition (EMD) and support vector regression (SVR). Experiments on the comparison dataset show that empirical mode decomposition combined with support vector regression can achieve performance close to empirical mode decomposition combined with support vector regression, when runtime takes less than one-fortieth of empirical mode decomposition combined with support vector regression (Yan and Chen 2017).

The question of whether the high volatility of prices could reflect market fundamentals has been widely discussed, although professional literature has confirmed the presence of price explosiveness in many financial markets. Using futures prices of six dominant nonferrous metals in the years 2014–2018, Ma and Xiong (2021) examined the characteristics and determinants of price explosiveness on futures markets of nonferrous metals. They found that, in the last 15 years, markets showed price explosiveness and that the ability of price mechanisms to adjust to price changes is limited.

Understanding metal prices enables producers, consumers, policy makers, and traders to predict short-term and long-term price trends and to prepare for them. Metal prices are interrelated because of their characteristics, as they are either jointly produced or consumed. While the effects of joint production of metals have recently been subject to analysis, the impacts of joint consumption on metal prices need to be systematically studied. Therefore, Shammugam et al. (2019) focus on analyzing the effects of joint consumption and joint production of metals on the causal relationship of metal prices.

The results of the study by Thazhugal Govindan Nair (2021) show that the movements of prices in metal markets are not random walk and current futures prices represent a reasonable estimate of the future "spot" metal prices. The author does not see any significant differences in the price efficiency in metal markets, which indicates the effect of limited idiosyncratic forces in the transmission of prices.

The importance of metal prices for the actual economic activity and financial markets has increased the focus on identifying price bubbles in precious and industrial metals. The study by Ozgur et al. (2021) analyzes monthly prices of gold, platinum, palladium, rhodium, silver and aluminum, copper, lead, nickel, steel, and tin over 1980M1-2019M12 and contributes to the existing literature by including generalized supremum augmented Dickey–Fuller (GSADF) test in the analysis to detect potential bubbles. The authors' findings suggest that financial factors play a crucial role in predicting metal price bubbles.

## 3. Materials and Methods

To achieve the goals of the paper, historical data on platinum futures prices will be collected from the website Investing [online], available at https://www.investing.com/ (accessed on 21 June 2022). A daily price of platinum in USD per troy ounce will be used. The data will include platinum prices for the period of 2 January 2017 to 11 March 2022 for each trading day.

**RQ1: Is platinum a store of value?**

Metal is considered a store of value if the movement of its price at least copies the value of the currency in which the investors hold their wealth. In the paper submitted, the currency is the Czech koruna. To be able to answer the research question, information on movements of the Czech koruna price is necessary. The price of the currency is also expressed in USD. This way enables avoiding the changes in the price levels of USD, which may affect the value of platinum. Data on the price of CZK will be obtained from the Czech

National Bank's exchange rate system available from the Czech National Bank [online], available from https://www.cnb.cz/cs/ (accessed on 21 June 2022).

To answer the research question, data for the period of 2 January 2017 to 11 March 2022 for each day the Czech National Bank announces the CZK–USD exchange rate are needed. The price of CZK expressed in USD will be rounded to 10 decimal points.

The relative price movement of both monitored assets, platinum and CZK, will be used to determine whether platinum is a store of value. The day-on-day movements of both prices will be observed and then mutually compared. In the event that the output is not clearly visible, linear regression will be used, which will then determine the trend of the difference of platinum and CZK prices.

**RQ2: Is platinum an investor's wealth multiplier?**

To be able to answer the second research question, the data on platinum and CZK prices for the time horizon of 2 January 2017–11 March 2022 will be used. However, in this case, the focus will be on the future trend of platinum and CZK prices instead of the past development, specifically to 31 December 2023. If the ratio in both assets' prices grows in favor of platinum, it can be stated that in the monitored period, platinum will be a wealth multiplier. For time-series smoothing and prediction of future development of both assets' prices, the recurrent neural network with long short-term memory layer (also referred to as LSTM) will be used. Both variables will use the same network architecture; however, the networks will be trained separately.

For both variables, an experiment will be carried out, aimed at determining the best parameters of the neural structure for each of them. The forecasting of the future platinum price will be performed accordingly using the methodology of Vochozka et al. (2021). The only exception is the overall structure and sequence of the layers of the neural network (for more details, see the diagram in Figure 1).

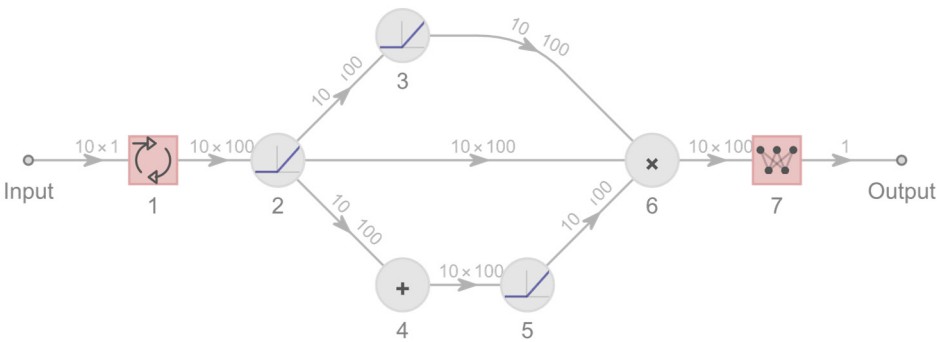

**Figure 1.** Architecture of artificial neural network with LSTM layer. Source: Author.

The neural network will consist of nine layers, with one input and one output layer and seven hidden layers:

1.  Input layer of neurons: a matrix that brings information on preceding prices of the commodity under review. As input information, the commodity price will be used (platinum or CZK) for the preceding 10 trading or announcement days. Therefore, the first information presented in the figure is $10 \times 1$.

2.  First hidden layer of neurons: it consists of the LSTM layer. LSTM as a neural network layer is specified separately below. It can even be considered an independent neural network embedded in another neural network (multilayer perceptron). The output of LSTM is a $10 \times n$ matrix (for illustration, the figure presents matrix $10 \times 100$). The number of elements in the output matrix is part of the experiment; the matrix size influences the predictive power of the model. A low number of elements can deteriorate the accuracy of the result; a high number can cause excessive complexity and overfitting of the model. In such a case, the model will show excellent parameters of trained neural network in time series smoothing, but the prediction of the future

development will be nonsensical. The number of elements will be determined at the interval of 5 to 2000 elements. The number of steps is determined at 1.

3.  Second hidden layer: it is the elementwise layer (a simple network that works with perceptron). Its purpose is to introduce nonlinearity into the neural network. Even in this case, a sub-experiment will be carried out and the suitability of the functions transmitting the signal will be tested. The function for this layer will be selected randomly:

    a.  Hyperbolic tangent (Tanh):

$$f(x) = \tanh x = \frac{e^x - e^{-x}}{e^x + e^{-x}}$$

    b.  Sinus (Sin):

$$f(x) = \sin x$$

    c.  Ramp (sometimes also ReLU):

$$R(x) : R \to R_0^+$$

    d.  Logistic function (logistic sigmoid):

$$f(x) = \frac{1}{1 + e^{-x}}$$

These four functions were selected based on long-term experiments, as they showed the best results (Vochozka et al. 2021).

For illustration, the Ramp function is used in Figure 1.

4.  Third hidden layers: elementwise layer. Its further setting corresponds to the second hidden layer (i.e., the specific function for transmitting the signal is selected randomly).
5.  Fourth hidden layer of neurons: a plus layer. Its task is to add signals of the second and the third hidden layers, unify, and send it on.
6.  Fifth hidden layer of neurons: elementwise layer. Its further setting corresponds to the second hidden layer (i.e., the specific function for transmitting the signal is selected randomly).
7.  Sixth hidden layer of neurons: a times layer. Its task is to multiply the signals of the third and the fifth layer and send it on.
8.  Seventh hidden layer of neurons: linear layer. At its input, there is a matrix of $10 \times n$ elements (in Figure 1, the matrix has $10 \times 100$ elements), and a vector with one element at its output.
9.  Output layer: predicted price of platinum or CZK (according to specific neural network).

*Long Short-Term Memory Layer*

The architecture of the artificial neural network is given in the methodology. However, the LSTM layer is a very specific neural network—a recurrent neural network.

The scheme of the LSTM layer assumes four basic processes (sub-networks) represented by input gate, output gate, forget gate, and memory gate. The state of the cell is defined as follows (Vochozka et al. 2021):

$$c_t = f_t * c_{t-1} + i_t * m_t$$

where

| | |
|---|---|
| $c_t$ | is a new state of the variable, |
| $f_t$ | is forget gate, |
| $c_{t-1}$ | is initial state of the variable, |
| $i_t$ | is input gate, and |
| $m_t$ | is memory gate. |

Input gate is defined as follows:

$$i_t = \sigma[W_{ix}x_t + W_{is}s_{t-1} + b_i]$$

where

$\sigma$　　is logistic sigmoid,
$W_{ix}$　is input weight in input gate, matrix $n \times k$,
$x_t$　　is input variable, matrix $n \times k$,
$W_{is}$　is weight in input gate, matrix $n \times n$,
$s_{t-1}$　is preceding state, and
$b_i$　　is bias, vector size $n$.

The state is defined by the equation below:

$$s_t = o_t * Tanh[c_t]$$

where

$s_t$　　is the state of the variable,
$o_t$　　is output gate, and
$Tanh$　is hyperbolic tangent.

Output gate is expressed as follows:

$$o_t = \sigma[W_{ax}x_t + W_{as}s_{t-1} + b_o]$$

where

$W_{ax}$　is the input weight in output gate, matrix $n \times k$,
$W_{as}$　is weight in the output gate, matrix $n \times n$, and
$b_o$　　is bias, vector size $n$.

An important innovation of LSTM is a forget gate:

$$f_t = \sigma\left[W_{fx}x_t + W_{fs}s_{t-1} + b_f\right]$$

where

$W_{fx}$　determines the input weight in forget gate, matrix $n \times k$,
$W_{fs}$　is weight in forget gate, matrix is $n \times n$, and
$b_f$　　is bias, vector size $n$.

In terms of the main processes, the last one to determine is memory gate:

$$m_t = Tanh[W_{mx}x_t + W_{ms}s_{t-1} + b_m]$$

where

$W_{mx}$　is the input weight in memory gate, matrix $n \times k$,
$W_{ms}$　is weight in memory gate, matrix $n \times n$, and
$B_m$　　is bias, vector size $n$.

Neural networks are trained using the ADAM algorithm, a method for stochastic optimization of neural network using adaptive moment estimation. For each commodity, the five best neural networks are selected, as some of them might suffer from overfitting. Therefore, five structures are analyzed so that the network that suffers from overfitting can be replaced by some other neural structure. The networks will be selected on the basis of the best performance of the network. This is measured using the Pearson correlation coefficient of smoothed time series and actual past development of the monitored assets' prices. Subsequently, the future trends of platinum and CZK prices are predicted using the best neural structure until 31 December 2023 (trading days only). If the difference of

both commodities' prices grows in favor of platinum, it can be stated that, in the monitored period, platinum was a wealth multiplier.

**RQ3: What are the properties of platinum as a precious metal?**

The data for answering this research question will be obtained using content analysis. A source of information will be university textbooks on the properties of platinum, its processing and use.

**RQ4: How is platinum traded on commodity markets?**

This question will also be answered using content analysis. The process will be analyses of a selected commodity stock market, (LME 2022). For the purposes of the paper, the focus is only on the important moments in the whole technology of trade.

## 4. Results

### 4.1. Platinum as a Store of Value

To determine whether the metal commodity is a store of value (or even a wealth multiplier), the trend of the time series, i.e., the price of platinum, shall be compared with the CZK price trend as a target asset in which the investors are assumed to hold their wealth. The development of platinum and CZK prices from the beginning of the year 2017 is presented in the graph in Figure 2.

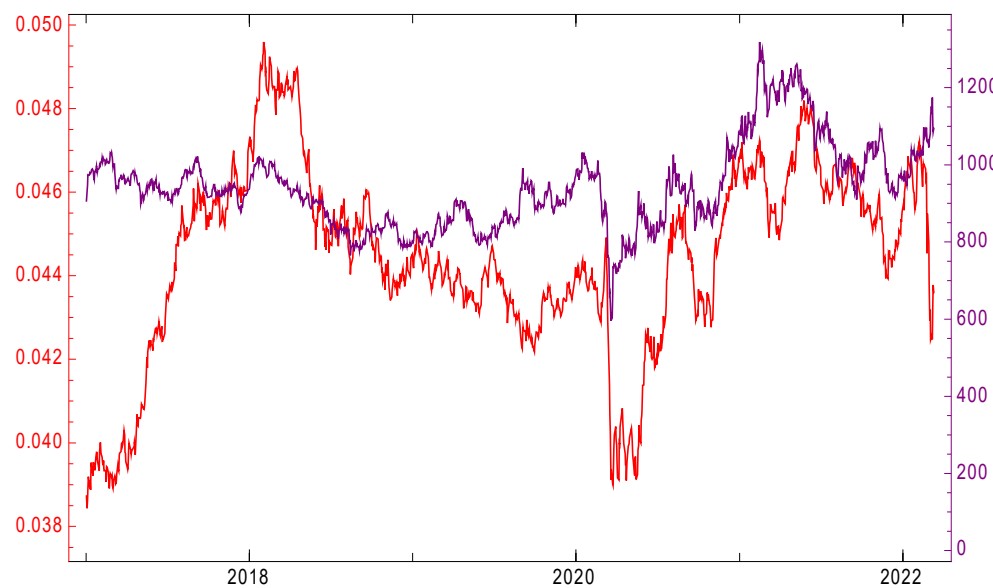

**Figure 2.** CZK and platinum prices in the years 2018–2022.

The two-sided chart shows the price of CZK and platinum in the years 2017–2022. The red curve represents the CZK price, while the violet curve is the price of platinum. The left vertical axis expresses the CZK price in USD, the right one the price for troy ounce of platinum in USD. The graph shows that the CZK price changed significantly in the monitored period, ranging from USD 0.0385 to 0.045 (i.e., by almost 15% in relative terms). The highest price of CZK was recorded in 2018, achieving the aforementioned USD 0.045. In 2019, the price decreased, achieving its minimum in 2020 (USD 0.0385). In the years 2021 and 2022, the CZK price grew. The price of platinum was very volatile. In the first years of the monitored period, the price was USD 800–1000 per troy ounce. In 2020, the price of the commodity dropped below USD 600, then rocketed to more than UD 1300 per troy ounce in the first quarter of the year 2022. The relative change in the price of the precious metal for a year and a quarter is thus more than 50%.

The day-on-day changes in prices are shown in Figure 3.

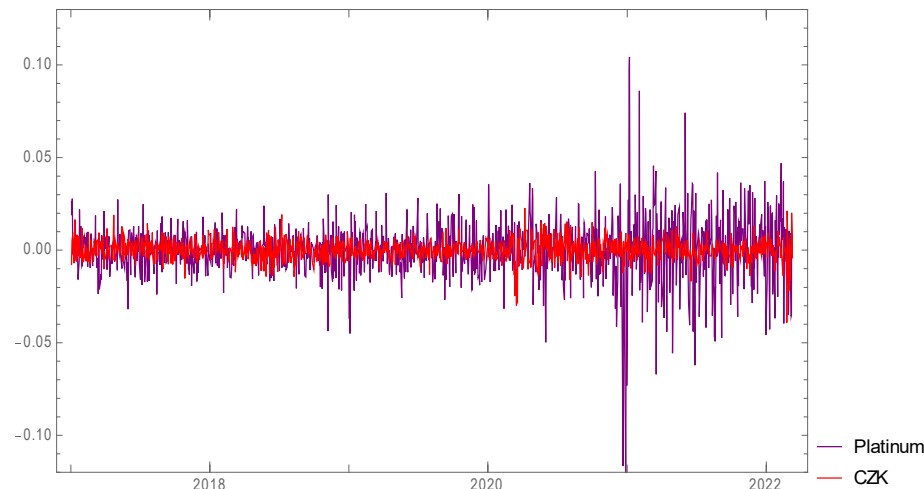

**Figure 3.** Differences of changes in CZK and platinum prices.

The graph shows the differences between the day-on-day changes in CZK and platinum prices expressed as an index. The changes in the commodities' prices are thus mutually comparable, although their absolute amounts are significantly different. The violet curve expresses the day-on-day differences of the platinum prices. The red curve represents the day-on-day differences in the CZK prices. The volatility of CZK is significantly lower than the platinum volatility in the period 2017–2022. The largest volatility was recorded at the beginning of 2021. It shall also be noted that platinum shows higher volatility from the year 2021 than in the preceding years of the monitored period. To compare the trends of both curves, the difference of day-on-day movements of both assets is calculated (Figure 4).

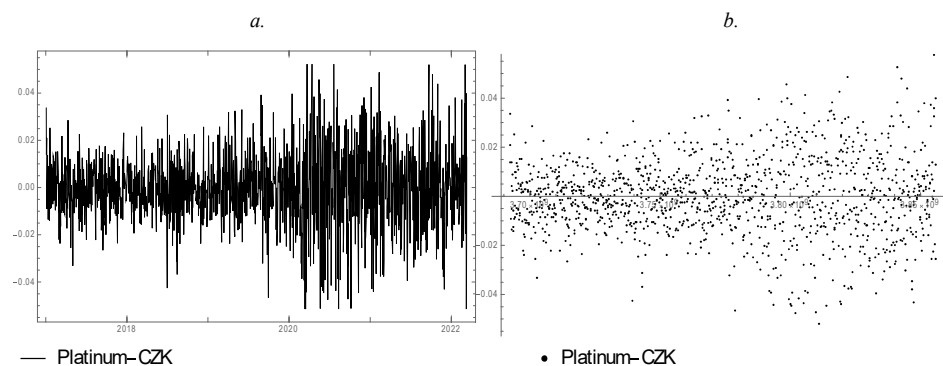

**Figure 4.** Difference of changes in platinum and CZK prices (**a**) Difference of changes in platinum and CZK prices as a line chart, (**b**) Difference of changes in platinum and CZK prices as a dot chart.

The figure shows the difference of day-on-day changes in the platinum and CZK prices. Figure 4a represents the difference of day-on-day changes in prices in the form of a line graph, and Figure 4b in the form of a scatter plot.

The figures indicate that the volatility of platinum grew faster than the CZK volatility. The difference in day-on-day changes is greater from the year 2020, not from the year 2021. To be able to determine whether platinum is a store of value, the direction of the regression curve of day-on-day value differences must be positive. The result of the regression analysis is shown in Figure 5.

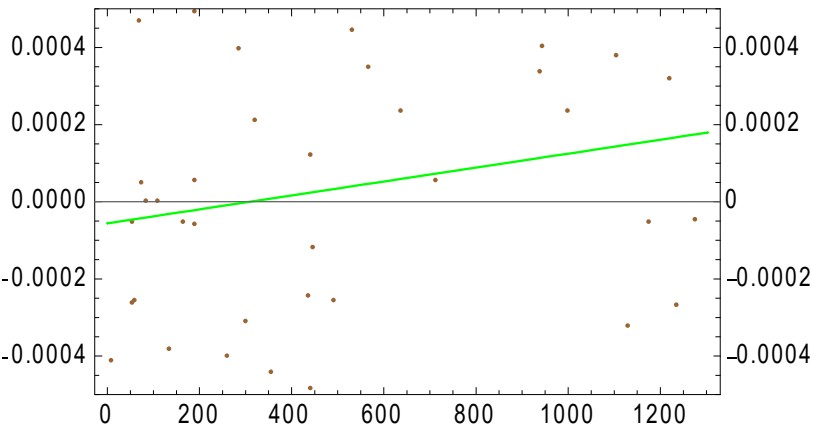

**Figure 5.** Difference in platinum and CZK price changes (detail).

Since the direction of the regression curve is too low, the output in Figure 5 is presented in a viewport determined within the interval of $(-0.0005, 0.0005)$. The magnification of the original graph is thus a hundredfold. The points in the graph represent the differences of day-on-day changes in platinum and CZK prices. The resulting regression curve is thus expressed as follows:

$$y = -0.00005583236162876401 + 1.80566725020571 * 10^{-7} * x$$

where

$y$      is the difference in day-on-day changes in platinum and CZK prices, and
$x$      is the number of time-series case.

The growth rate of the price of platinum is $1.80566725020571 \times 10^{-7}$ higher than the growth rate of the price of CZK. It can thus be concluded that platinum is a store of value for a koruna investor (from the perspective of long-term investment).

### 4.2. Platinum as a Multiplier of Investor Wealth

The above results and mainly the direction of the regression curve indicate that platinum is, or rather was, a wealth multiplier in the monitored period. To confirm the future validity of this statement, it is necessary to determine the further development of platinum and CZK prices and, based on the difference of directions, to estimate whether the price of platinum will grow faster than the price of CZK to USD.

#### 4.2.1. Prediction of Platinum Price

A part of the research was the generation of neural networks. The overview of the five best-retained networks is presented in Table 1.

**Table 1.** Retained neural networks for predicting platinum prices.

| ID of Neural Networks | Recurrent Layer | Number of Outputs of the 1st Layer | Function of the 3rd Layer | Function of the 4th Layer | Function of the 6th Layer | 8th Layer |
|---|---|---|---|---|---|---|
| 1-NN_PLT | LTSM | 283 | Sin | Logistic Sigmoid | Logistic Sigmoid | Linear |
| 2-NN_PLT | LTSM | 148 | Tanh | Logistic Sigmoid | Logistic Sigmoid | Linear |
| 3-NN_PLT | LTSM | 224 | Tanh | Logistic Sigmoid | Logistic Sigmoid | Linear |
| 4-NN_PLT | LTSM | 264 | Ramp | Logistic Sigmoid | Logistic Sigmoid | Linear |
| 5-NN_PLT | LTSM | 115 | Tanh | Logistic Sigmoid | Logistic Sigmoid | Linear |

Source: Authors.

The architecture of the neural networks is determined by the methodology. The first hidden layer (the second layer in general) of the neural network consists of LTSM. At the

output of the first hidden layer, there is the m × n matrix, where m = 10 (number of input data) and n is the number of outputs of the first layer in the table. In the case of retained neural networks, n ranges from 115 to 283. In the third layer, the neural networks differ in the way the signal propagates. The functions used are sine, hyperbolic tangent, and ramp. The best neural networks use the logistic function in the fourth and sixth layer to propagate the signal. The eighth layer is a linear layer in all neural networks (following the assumed architecture). The smoothed time series according to the retained (the best) neural networks is presented in Figure 6.

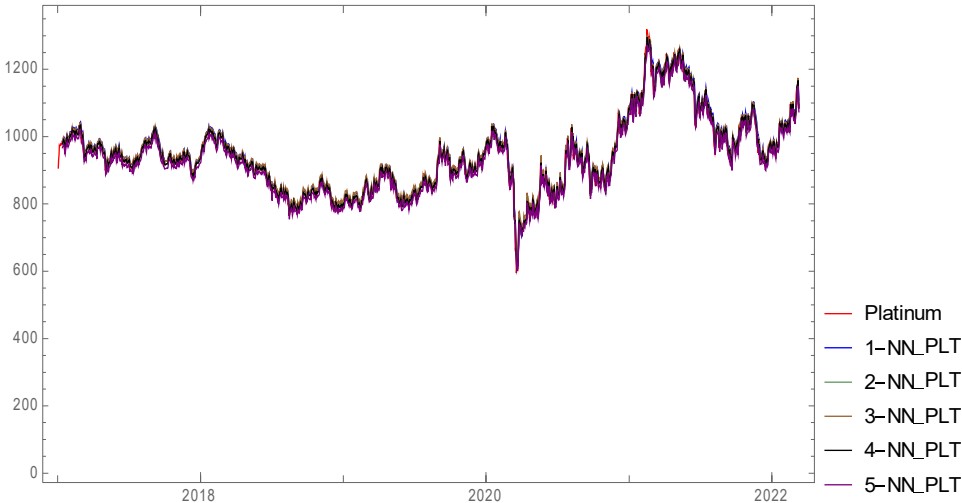

**Figure 6.** The current trend of platinum price in USD and smoothed time series.

The figure shows the current trade of platinum price in USD (the red curve) and smoothed time series, the models (other colors). The smoothed time series are calculated using trained neural networks. The smoothed time series successfully copy the trend of platinum price in USD. In the year 2018, the price of platinum was approx. USD 1000. In the next year, the price of platinum and smoothed time series achieved USD 800. In the year 2020, the price decreased to USD 600, which was followed by growth. In 2021, the price of platinum and smoothed time series was USD 1000. In 2022, it decreased to USD 900. The statistics of smoothed time series are presented in Table 2.

**Table 2.** Statistical characteristics of smoothed time series and platinum prices.

| Time Series | Minimum | Maximum | Mean | Standard Deviation | Variance |
|---|---|---|---|---|---|
| Platinum | 595.2 | 1318.75 | 940.86 | 110.11 | 12,124.58 |
| 1—NN_PLT | 636.45 | 1285.08 | 950.09 | 111.96 | 12,534.69 |
| 2—NN_PLT | 620. 53 | 1296.63 | 941.22 | 108.52 | 11,776.38 |
| 3—NN_PLT | 625.91 | 1302.66 | 950.65 | 108.92 | 11,863.35 |
| 4—NN_PLT | 601.83 | 1295. 91 | 941 | 110.72 | 12,258.4 |
| 5—NN_PLT | 604.75 | 1271.81 | 928.57 | 109.54 | 11,999.44 |

Source: Authors.

The minimum value of platinum (595.2) is smaller than in the case of other smoothed time series. Similarly, the maximum is also higher than in the case of other smoothed time series, achieving the value of 1318.75. The smoothed time series 3 achieves the maximum of 1302.66, which is a value closest to the maximum value of platinum. The mean value of platinum is 940.86. The table shows that the highest mean value is achieved by the smoothed time series 3. The value of standard deviation of platinum is lower compared to other smoothed time series. The variance in the case of platinum is 12,124.58, which

means that it is lower compared to other time series. Figure 7 shows the historical prices and the possible development of platinum price until the end of the year 2023 by the individual models.

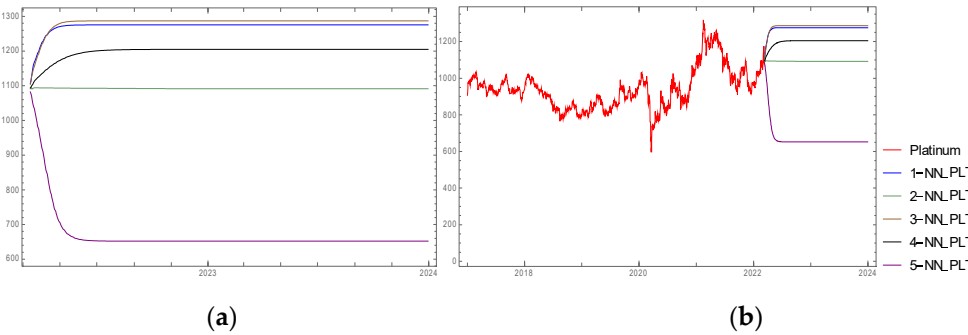

(**a**)          (**b**)

**Figure 7.** Prediction of future development of platinum prices till the end of the year 2023 (**a**) future development of platinum price according to the best neural networks, (**b**) past development of platinum price with its future development according to the best neural networks.

The neural network assumes a decline in price to USD 650 per troy ounce in one case. 2-NN_PLT assumes a constant price of USD 1000 per troy ounce. The remaining three networks present the growth of platinum price until the end of the year 2023, where the price might achieve USD 1200, 1275, or 1300 per troy ounce. The continuity of the historical prices and predicted future prices of platinum are presented in the graph in Figure 7. The statistics of the predictions with the historical prices are shown in detail in Table 3.

**Table 3.** Statistical characteristics of historical time series and predicted platinum prices.

| Time Series | Minimum | Maximum | Mean | Standard Deviation | Variance |
|---|---|---|---|---|---|
| Platinum | 1106.82 | 1275.81 | 1271.63 | 19.27 | 371.51 |
| 1—NN_PLT | 1091.17 | 1094.67 | 1092.03 | 0.78 | 0.61 |
| 2—NN_PLT | 1103.92 | 1287.27 | 1282.12 | 22.56 | 508.90 |
| 3—NN_PLT | 1094.28 | 1205.06 | 1198.63 | 18.61 | 346.30 |
| 4—NN_PLT | 652.2 | 1082.67 | 670.88 | 69.62 | 4846.41 |

Source: Authors.

Table 2 presents the statistics of platinum prices predictions and smoothed time series. The minimum platinum price is USD 1106.82, which means that the price is higher than in the case of other smoothed time series. The maximum price is USD 1275.81, i.e., a lower value compared to other smoothed time series. The average price of platinum is USD 1271.63, a lower value than in the case of other smoothed time series. The same applies for the standard deviation (USD 19.27), which is also lower than in the case of other smoothed time series. Similarly, the value of variance is lower in the case of platinum, achieving the value of USD 371.51. This is a significant difference, as the value of variance for 4—NN_PLT is 4846.41.

The most successful neural structure is 1—NN_PLT. This is given by combining statistical characteristics and a real estimate of platinum prices (the predicted development is shown in Figure 8).

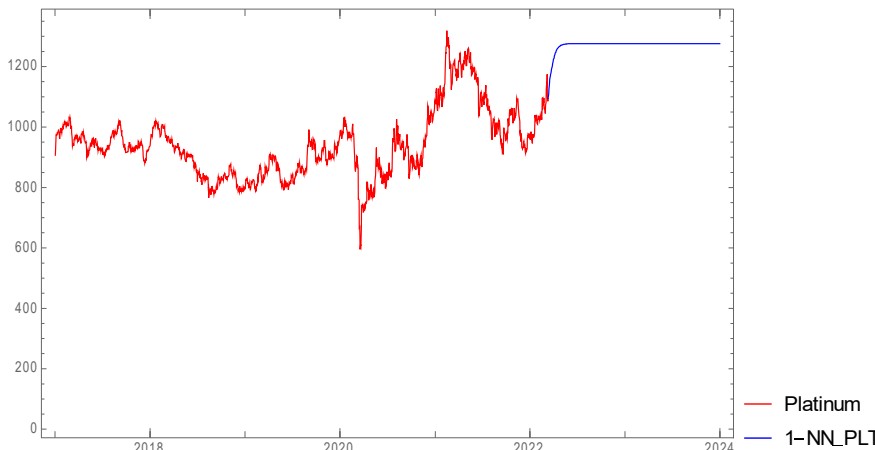

**Figure 8.** Past development of platinum prices from 2017 and predicted prices until the end of the year 2023.

The graph shows the past development of platinum prices from the year 2017 and the prediction until the year 2023. The selected time series is 1—NN_PLT. It follows from the graph that the development of the price of platinum was relatively stable until 2020. At the beginning of 2020, the price dropped to USD 600 per troy ounce. In contrast, in 2021, it achieved its highest values (USD 1300). The prediction until 2023 suggests fast growth to approx. USD 1275 per troy ounce and stabilization at this value.

### 4.2.2. Prediction of CZK Price

In the case of predicting CZK price, analogous development is expected. Table 4 presents the characteristics of the retained neural networks (with the best performance).

**Table 4.** Retained neural networks for predicting CZK price.

| ID Neural Networks | Recurrent Layer | Number of Outputs of the 1st Layer | Function of the 3rd Layer | Function of the 4th Layer | Function of 6th Layer | 8th Layer |
|---|---|---|---|---|---|---|
| 1—NN_CZK | LTSM | 472 | Ramp | Sin | Sin | Linear |
| 2—NN_CZK | LTSM | 98 | Tanh | Logistic Sigmoid | Logistic Sigmoid | Linear |
| 3—NN_CZK | LTSM | 105 | Ramp | Ramp | Tanh | Linear |
| 4—NN_CZK | LTSM | 114 | Sin | Logistic Sigmoid | Logistic Sigmoid | Linear |
| 5—NN_CZK | LTSM | 71 | Ramp | Sin | Tanh | Linear |

Source: Authors.

The first hidden (the second layer in general) layer of the neural network consists of LTSM. At the output of the first hidden layer, there is matrix m × n, where m = 10 (the number of the input data) and n is a number of outputs of the first layer in the table, ranging from 71 to 472 in the retained neural networks. In the third layer, neural networks differ from each other by the way of transmitting the signal, using the sine, hyperbolic tangent, and ramp functions. In the fourth layer, the signal transmits using the sine, ramp, and logistic functions. The functions of the sixth layer are sine, hyperbolic tangent, and logistic functions. The eighth layer consists of a linear layer in all neural networks (in line with the assumed architecture). The smoothed time series by retained (best) neural structures is shown in Figure 9.

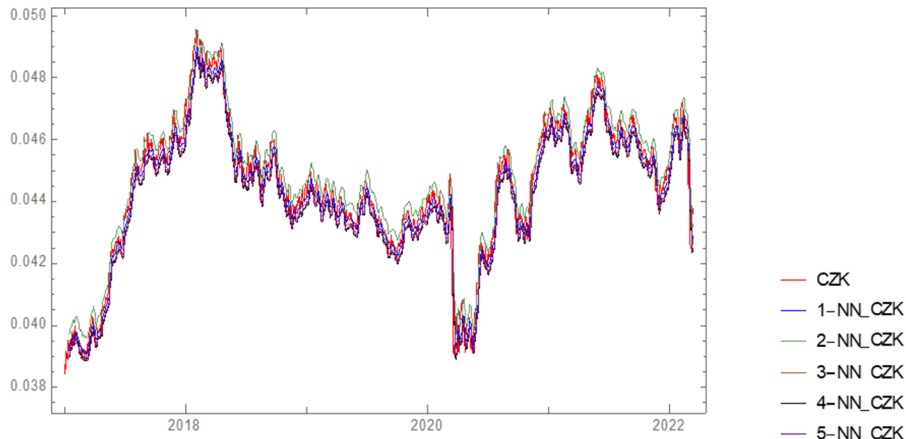

**Figure 9.** The current trend of smoothed time series and CZK price in USD.

The figure shows the current trend of CZK price in USD (the red curve) and smoothed time series (models) (other colors). The smoothed time series very precisely copy (with a minimum deviation) the real trend of the price of the monitored metal commodity. The statistics of the real time series "CZK" and smoothed time series of neural networks include minimum, maximum, mean, standard deviation, and variance (see Table 5).

**Table 5.** Statistical characteristics of smoothed time series and CZK price.

| Time Series | Minimum | Maximum | Mean | Standard Deviation |
|:---:|:---:|:---:|:---:|:---:|
| CZK | 0.0384 | 0.0496 | 0.0444 | 0.0023 |
| 1—NN_CZK | 0.0390 | 0.0490 | 0.0443 | 0.0021 |
| 2—NN_CZK | 0.0396 | 0.0496 | 0.0449 | 0.0021 |
| 3—NN_CZK | 0.0389 | 0.0488 | 0.0442 | 0.0021 |
| 4—NN_CZK | 0.0389 | 0.0488 | 0.0441 | 0.0021 |
| 5—NN_CZK | 0.0389 | 0.0488 | 0.0441 | 0.0021 |

Source: Authors.

The minimum of the actual CZK time series and smoothed time series is approximately the same, achieving the value of 0.04. The maximum value is 0.05, which is nearly the same as in the case of the smoothed time series. The mean value of the real CZK time series and smoothed time series is about 0.044. The standard deviation of the real CZK time series and smoothed time series ranges from 0.0021 to 0.0023. Figure 10 presents the historical prices and the possible (predicted) development of platinum prices until the end of 2023 by individual models.

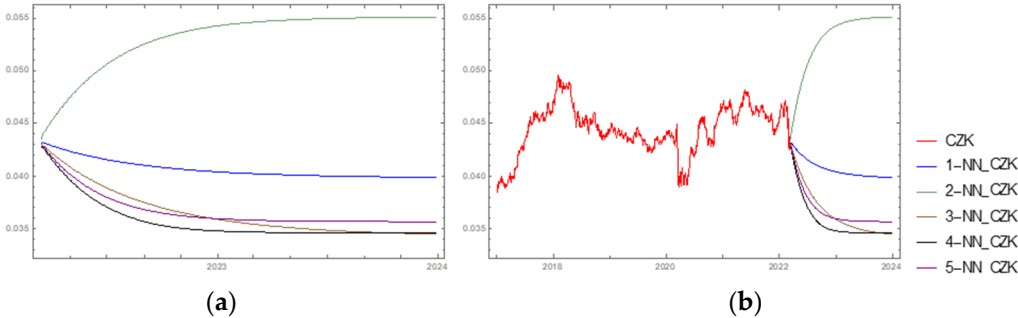

(a)  (b)

**Figure 10.** Prediction of the future trend of CZK price until the end of 2023 (**a**) future development of CZK price according to the best neural networks, (**b**) past development of CZK price with its future development according to the best neural networks.

The neural network assumes the growth of CZK price to the value of USD 0.055 in one case. Other models predict a decline in the CZK price until the end of the year 2023. However, the price may achieve the value of USD 0.035–0.04 per one CZK. The continuity of the historical price and the predicted future price of platinum are presented in the graph in Figure 10b. The statistics of the predictions with the historical priced are described in detail in Table 6.

**Table 6.** Statistical characteristics of historical time series and predicted CZK prices.

| Time Series | Minimum | Maximum | Mean | Standard Deviation |
|:-----------:|:-------:|:-------:|:----:|:------------------:|
| CZK | 0.0399 | 0.0432 | 0.0406 | 0.0009 |
| 1—NN_CZK | 0.0444 | 0.0551 | 0.0531 | 0.0028 |
| 2—NN_CZK | 0.0345 | 0.0430 | 0.0365 | 0.0022 |
| 3—NN_CZK | 0.0346 | 0.0428 | 0.0357 | 0.0019 |
| 4—NN_CZK | 0.0357 | 0.0428 | 0.0367 | 0.0017 |

Source: Author.

The table presents the statistics of CZK price predictions and smoothed time series when comparing the minimum, maximum, mean, standard deviation, and variance. The minimum price of CZK the same or higher compared to the smoothed time series, achieving approx. USD 0.04; the maximum price is the same or lower compared to smoothed time series, with the values ranging between USD 0.043 and 0.055. The mean value of the CZK price ranges between USD 0.0357 and 0.0531. The standard deviations of both time series achieve the values of USD 0.0009–0.0028. The most accurate prediction is thus the prediction by 1—NN_CZK (on the basis of the statistical characteristics of the smoothed time series and possible development of CZK price).

The Figure 11 shows the past development of the CZK price from 2017 and the prediction until the end of 2023. In 2017, the CZK price was USD 0.038. Subsequently, it grew to USD 0.050, which was followed by a decline in 2019. In 2020, the price achieved USD 0.038 and then grew again. In 2022, it was USD 0.047. In the case of the smoothed time series 1—NN-CZK, the predicted trend until the end of 2023 is downward, achieving the value of USD 0.04 per one CZK.

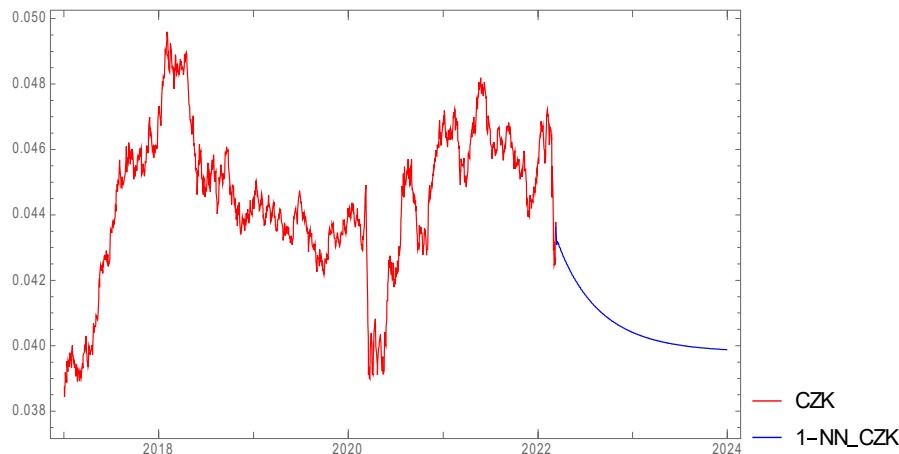

**Figure 11.** Past development of CZK price from 2017 and the prediction until the end of 2023.

4.2.3. Summary

The basic data for RQ 2 are presented in Figure 12a,b.

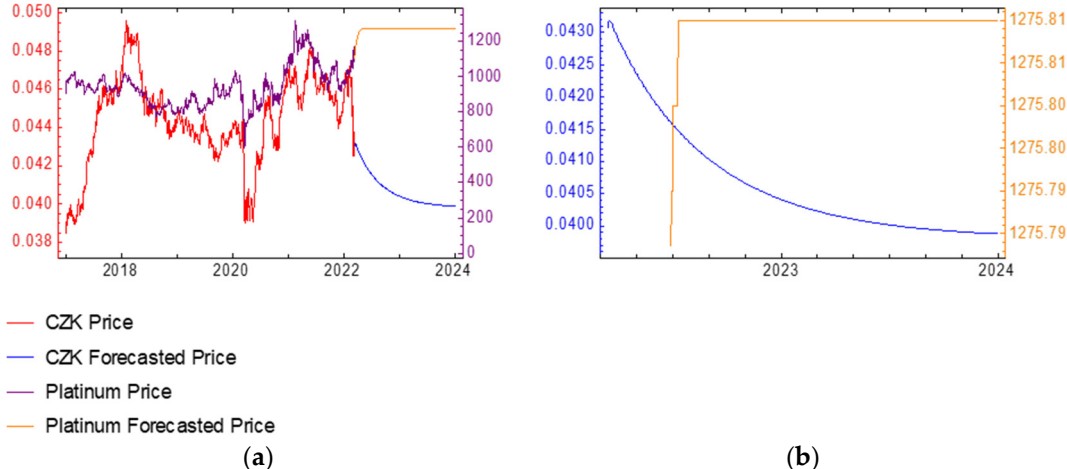

(**a**)  (**b**)

**Figure 12.** Past development of platinum and CZK prices and their prediction until the end of 2023 (**a**) past development of platinum and CZK prices with their future development to the end of 2023, (**b**) cutout of the future development of platinum and CZK prices.

The combined graph in Figure 12a shows the development of prices of both time series (platinum and CZK prices expressed in USD in both cases) with the predicted values until the end of 2023. The red curve represents the CZK price, the blue one the future price of CZK, the violet curve the platinum price, and the orange one the future platinum price. The left vertical axis shows the prices of CZK, the right one the platinum prices. The combined graph in Figure 12b shows the future development of CZK prices and partly the prediction of the platinum prices until the end of 2023.

The figure clearly shows the directions of both curves. It can thus be concluded that for a koruna investor, platinum will be a wealth multiplier (at least until the end of 2023).

### 4.3. Properties of Platinum as a Precious Metal

Platinum (Pt) is a malleable, ductile, silver-white metal, with the atomic number 78 and an atomic weight of 195.09. In nature, it occurs in a metallic state, alloyed with a small amount of iron, copper, other platinum metals, and sometimes also with a small amount of gold (Populeanu et al. 2018). Platinum is a chemical element, a precious metal used in industry. As the rarest elements in the lithosphere, metals of the platinum group are technological actors in many modern industrial processes, especially automotive. (Xun et al. 2020). Platinum and platinum-based materials with high catalytic performance and chemical and mechanical stability are of crucial importance for electronic devices, biomedicine, optics, oil, and automotives (Akbar et al. 2018).

Rhodium, ruthenium, palladium, and platinum are classified as platinum group metals (PGM). Their occurrence in nature is rare; therefore, it is necessary both from the economic and ecological point of view to develop an effective process of obtaining PGM from waste/secondary resources, such as used car catalytic converters (Rzelewska and Regel-Rosocka 2018).

Platinum is also one of the elements with the highest density; its density is 12.4 oz/sq.in (21.45 g/cm$^3$), which is slightly more than 21 times higher than the density of water or 6 times higher than the density of diamond, according to Chemicool. These properties enable a wide range of application of this very rare and precious metal (Live Science 2016). Morphologic properties of platinum coatings are influenced by the source state and by the sputtered particles energy (Mroz et al. 2019).

*4.4. Platinum Trading on Commodity Markets*

Platinum and palladium auctions take place twice a day, at 9:45 a.m. and 2 p.m. (London time). The auction of palladiums starts after the price of platinum is determined (LME 2022).

LME auctions are controlled so that they can proceed only if a sufficient number of participants are present (the Quorum). When the price is confirmed and trades allocated, the price is announced via LME.com and data distributors. All platinum and palladium transactions are Loco London and are settled on a bilateral basis.

LME bullion provides fully automated auctions with displayed prices. When a potential execution price is displayed, house traders, client traders, and direct clients of participants are asked to express their interest. If the interest of all participants is in the permitted tolerance, the price is confirmed and subsequent trade auctions are redistributed to the participants. If the deviation is not in the allowed tolerance, a new price is calculated and displayed to the participants so that the next round can start.

The prices are announced twice a day, shortly after their confirmation. This includes morning and afternoon outright platinum and palladium prices in USD, the volumes offered and required, and a total imbalance. The prices are per troy ounce.

The live auction website provides real-time updates of auctions, prices being tried, and total volumes. Monthly averages for the LBMA platinum price and LBMA palladium price are announced on the last trading day of each month.

LME administers the licensing of prices of LBMA platinum and LBMA palladium for companies that want to redistribute, to use data as a reference in transactions, for pricing and valuation services, or for creating derived products, such as indexes and ETF.

The delivery payable in accordance with the client metal contract is made on the date of the call by delivering warrants. Warrants are delivered by means of transfer as defined by the LME and in accordance with the regulations and operational procedures for LMES, which govern ex-cleared transfers. Warrant weights are accepted in all cases between the buyer and seller.

Warrants must be issued by a stock listed on the stock exchange and must have properties determined by the special contract rules for the relevant metal.

## 5. Discussion

*RQ1: Is platinum a store of value?*

The results show that platinum is a store of value for a koruna investor (in terms of long-term investment). The development of platinum prices was examined over the time horizon of five years, considering all trading days from the beginning of 2017 until the beginning of 2022. Moreover, the authors examined the development of CZK prices in the same period. A model of behavior of both time series was created. Subsequently, further development of both investment commodities until the end of 2023 was predicted. The models predict a slight decrease in CZK prices and fast growth and stability for platinum prices. Given that imaginary scissors of both assets open over time, it can be stated that it pays koruna investors to store their wealth in the form of an expensive commodity—platinum.

*RQ2: Is platinum a multiplier of investor's wealth?*

When confirming that platinum is a store of value, and thus the wealth of investors, the focus is on platinum as a potential multiplier of investor's wealth. The answer to RQ1 confirmed that platinum stores a real value of the investor's wealth. The second research question examines whether investment in platinum increases the real value of a koruna investor's wealth. For this purpose, the directions of the two curves, specifically the curve of the predicted future prices of platinum in USD and prices of CZK in USD, are also compared. The more progressively the direction of the curve representing the platinum price grows compared to the direction of the curve of CZK prices in USD, the greater the increase in the real value of a koruna investor's wealth. In this case, it is clear at first glance. The direction of the CZK price curve decreases, while the direction of the platinum price shows an upward trend. However, due to the negative direction of CZK, another condition

must be set so that it can be confirmed that platinum is a wealth multiplier; specifically, the direction of the regression curve of platinum must be positive. Both conditions were thus fulfilled. The direction of the regression curve of the predicted platinum price is positive, while the direction of the predicted CZK price is negative. It can thus be concluded that platinum is a store of value, as well as a multiplier of a koruna investor's wealth.

*RQ3: What are the properties of platinum as precious metal?*

Platinum (Pt) is a precious metal, which is malleable, ductile, and has a silver-white color. As specified above, its atomic number is 78 and atomic weight is 195.09. It is also one of the elements with the highest density (12.4 oz/sq.in.–21.45 g/cm$^3$), which is slightly more than 21 times higher than the density of water or 6 times higher than the density of diamond, according to Chemicool. These properties enable a wide range of applications of this very rare and precious metal (Live Science 2016). An even more important property is that platinum can be stored in the physical form. It is separable and, because of its scarcity, it can accumulate great value. It can be accumulated, is durable, and does not lose its value when stored.

*RQ4: How is platinum traded on commodity markets?*

RQ4 is aimed at determining whether the real value of a precious metal is traded on commodity exchanges or whether it is only trust that the warranty (which bears almost no value in reality) is exchangeable for real value. Trade in commodities in the form of precious metals is based on trust between a seller and a buyer. The sellers believe that they will receive some other asset bearing value (money) for the commodity; buyers believe that they obtain assets that are bearers of value (platinum in this case). However, in fact, what is traded are warranties, which are certificates exchangeable for the commodity that is the subject of the trade, while the precious metal is stored in a special deposit. After the trade is executed, the warrant is either further traded or exchanged for a precious metal (especially in the case of short-term trade).

Platinum auctions take place twice a day, from 9:45 a.m. and from 2 p.m. (London time). When the platinum price is determined, the auction of palladium starts (LME 2022). All platinum transactions are Loco London and are settled on a bilateral basis.

Prices are announced twice a day, shortly after their confirmation, and include both morning and afternoon outright prices of palladium and platinum in USD, required and offered volumes, and a total imbalance. The prices are per troy ounce.

The delivery payable in accordance with the client metal contract is made on the date of the call by delivering warranties. Warranties are delivered by transfer as defined by the London Metal Exchange in accordance with the regulations and operational procedures for the LMES (London Metal Exchange Securities), which governs ex-cleared transactions. Warranty weights are accepted in all cases by buyers and sellers. Warranties must be issued by a stock listed on the stock exchange and must have features prescribed by the special contract rules for a relevant metal.

It follows that on the stock exchange, an asset that bears a value—platinum (because of its possible use, physical and chemical properties, and scarcity) is traded on one side and, on the other side, an asset whose value is based on the trust of an individual that they will receive real value (money) for the asset.

## 6. Conclusions

The goal of the paper was to determine whether platinum can be a suitable investment. For this purpose, content analysis and the method of multiplayer perceptron neural networks were used. Based on the results, it can be stated that it pays for a koruna investor to keep their wealth in the form of an expensive commodity—platinum.

To be able to achieve the set goal, it was necessary to react, to look at the problem from different perspectives. As a potential investor, a person or company that has allocated their wealth in CZK was selected. CZK is the currency of the Czech Republic, which is a relatively small country. CZK has a flexible exchange rate determined on the basis of the supply and demand for this currency. Since it is a national currency, it is regulated by a

central bank, which works actively with the reserves of commercial banks, creating foreign exchange reserves used for interventions in international currency markets, and uses a wide range of other tools to reduce the volatility of the CZK in a free convertibility measure. In this sense, to achieve the research goal it was necessary to determine whether platinum is a store of value for a koruna investor. Therefore, a regression was performed, when a direction of the curve was sought that characterizes the past and future development of both commodities, i.e., platinum and CZK. The result opens imaginary scissors between the future development of platinum and CZK prices (both expressed in USD). It can thus be concluded that platinum is a store of value but, from the long-term perspective, it is also a multiplier of a koruna investor's wealth.

The other two research questions examined whether investors gain real value by buying platinum and whether the market is liquid enough. Both questions were examined in the environment of the London Metal Exchange. The first question to be addressed concerned the store of value. If an investor has cash in any currency, when moving away from the golden standard, the individual does not hold any specific value, only the trust that it will be possible to buy a valuable asset for money. Money trust must be universal so that it could be stated that money has real value. Therefore, it was examined whether the investor has a valuable asset when the trade is executed or just a certificate and trust that this certificate is convertible into a valuable asset. On the LME, an investor trades and can exchange the certificate for a specific amount of precious metal at the LME contract warehouse. When considering the complex logistics of a potential trade in physical units of metal, it can be stated briefly that the investor becomes an owner of a valuable asset. Moreover, as the asset is valuable, the stock exchange is always willing to trade the metal further, if the basic rules are accomplished. This means that in these markets, there is always a buyer who wants to buy the precious metal at a certain price (the price refers to an equivalent of a freely convertible world currency). Another prerequisite for achieving the research goal are the properties of the traded metal. The properties of the metal under review are important for any investment: it is separable, it can be accumulated, and its physical and chemical properties are not lost by its storage. It can be stored relatively easily.

It can thus be concluded that platinum is a store of value, as well as a wealth multiplier for a koruna investor.

A research limitation is the period for which the data used were obtained. Therefore, it shall be specified that platinum is a store of value but also a wealth multiplier when using data for a five-year period. At the same time, it shall be added that for this period, no turbulent changes (interruptions in platinum supply, unexpected government regulation of trade, etc.) have been considered. At the same time, it is interesting to observe the individual parameters of generating value. Although it was stated in the text that a model based on individual parameters of platinum price formation could be very complicated, an interesting direction is to divide the price formation parameters into financial and non-financial. Maseko and Musingwini (2019) concluded that non-financial parameters influence platinum prices by about 5%. The question is whether this percentage is correctly determined, as Maghyereh and Abdoh (2022) state that sentiment creates predictive bubbles that affect the price of precious metals, including platinum, and this precious metal is often overvalued. Predictive bubbles thus represent a very interesting future research topic. Other authors such as Bekiros et al. (2017) or Akhtaruzzaman et al. (2021a) explored the role of precious metal during extreme situations. They also point out the role of gold as an interesting investment asset. Other authors such as Akhtaruzzaman et al. (2021b) and Boubaker et al. (2022) examined or at least mentioned the possibility of the investment potential of pairs of precious metals. So, the pair in the shape of gold–platinum can be a suitable investment for real store of wealth in extreme situation (like the COVID-19 pandemic or war). Pairs of precious metals thus represent a very interesting future research topic.

**Author Contributions:** Conceptualization, M.V. and Z.R.; methodology, M.V.; software, M.V.; validation, Z.R. and A.B.; formal analysis, A.B.; investigation, M.V.; resources, A.B.; data curation, Z.R.; writing—original draft preparation, A.B.; writing—review and editing, A.B.; visualization, M.V.; supervision, M.V.; project administration, Z.R.; funding acquisition, M.V. All authors have read and agreed to the published version of the manuscript.

**Funding:** This research was funded by Institute of Technology and Business in České Budějovice grant number 07SVV22.

**Institutional Review Board Statement:** Not applicable.

**Informed Consent Statement:** Not applicable.

**Data Availability Statement:** Data available upon request.

**Conflicts of Interest:** The authors declare no conflict of interest.

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
