# Peer review of "Is Platinum a Real Store of Wealth?"

_ijfs, doi:10.3390/ijfs10030070_

Round 1
Reviewer 1 Report
Let me start with listing the main strength of the paper “Is Platinum a Real Store of Wealth?”. The main one is the topic itself.
The study put forward by the researchers is interesting and the structure of the document is valid. The methodology used allows meeting the proposed objective. Although, as the authors point out, the main research limitation is the period for which the data were used.
However, there are several aspects that are unclear to me.
First, what drives platinum value? Like all commodities, it is a complex mix of supply and demand fundamentals.
Also, as a result of commodities shock due to the rise of those products and supply chain disruption some attention to this should be carried out.
There is a lack of research about those previous ideas
Also, the literature review should be updated and the conclusions and future research should be rewritten in order to highlight the importance of the paper.
Besides the discussion and conclusions should be rewritten.
Therefore, I consider that a very deep major revision should be done by authors.
Author Response
Dear reviewer,
Thank you very much for your valuable comments. We have made major changes of our paper. We have changes of all parts of the paper. We have asked native speaker for proofreading, literature review have been updated (some irrelevant sources were erased, some now added). We have changed presentation of the results of our research. We have made even other changes (presentation of methodology, discussion and conclusion).
We hope you will find our changes valuable.
With Best Regards
Marek Vochozka
Reviewer 2 Report
Dear authors,
Thank you very much for sending your paper to the journal. However, based on my following comments the paper is rejected at this stage:
1-lack of originality and contributions
2- Underdevelop theoretical issues
3- Unrelated literature
4- Vague research methodology
Author Response
Dear reviewer,
Thank you very much for your valuable comments. We have made major changes of our paper. We have changes of all parts of the paper. We have asked native speaker for proofreading, literature review have been updated (some irrelevant sources were erased, some now added). I hope we have better explain the originality of the research. We have changed presentation of the results of our research. We have made even other changes (presentation of methodology, discussion, and conclusion).
We hope you will find our changes valuable.
With Best Regards
Marek Vochozka
Reviewer 3 Report
I congratulate the authors for a comprehensive and well-developed research in a very interesting domain.
The research is well conducted with an introduction covering the main issues of the study, a detailed literature review, methods and methodology fine grounded and proving the results in research questions.
Discussions are guided and tracked to provide detailed answers to each question introduced in the study.
Author Response
Dear reviewer,
Thank you very much for your valuable comments. We have asked native speaker for proofreading. We have made other changes according to the comment of other reviewers.
We hope you will find our changes valuable.
With Best Regards
Marek Vochozka
Round 2
Reviewer 1 Report
I am glad to see the authors have made several changes.
I have no further comments.
Author Response
Dear Reviewer,
Thank you very much for your positive opinion. The paper was corrected by the native speaker (professional proofreader).
With Best Regards
Marek Vochozka
Reviewer 2 Report
Dear authors,
Thank you very much for sending the revised paper. You did not incorporate all my comments into the paper. My main concerns were as follows:
1- What is the research gap?
2- What is the research originality?
3- What are the research problem(s)?
4- You need to conduct robustness test to validate the primary findings
5- It is recommended to conduct proofreading on the paper
Author Response
Dear reviewer,
Thank you for your notes. However, we must admit that I do not understand them. They are too vague for us. Just in short:
Research gap was described in the Introduction.
Originality was cleared and proofed by citations in Introduction (and in the reaction to the Introduction – in Discussion and Conclusion).
The research questions were defined in the Introduction.
I think that the reviewer is not expert in neural networks (and maybe in statistics at all). The time series are continuous and long lasting.
We have ordered proof-reading by native professional speaker. So, I do not know what to do more about it (maybe other and other native speaker can do it). Maybe I can send a certificate.
The other reviewers accepted the paper after revisions.
To sum up, we would like to answer your note, however we do not understand them. So, the paper was now without any changes.
If you insist on the corrections, please provide us better explanation, or advise us to withdraw the paper.
With Best Regards
Marek Vochozka